DOI: 10.1038/s41467-018-05588-5 | OPEN

# PARP2 mediates branched poly ADP-ribosylation in response to DNA damage

Qian Chen[1], Muzaffer Ahmad Kassab[1], Françoise Dantzer[2] & Xiaochun Yu[1]

Poly(ADP-ribosyl)ation (PARylation) is a posttranslational modification involved in multiple biological processes, including DNA damage repair. This modification is catalyzed by poly (ADP-ribose) polymerase (PARP) family of enzymes. PARylation is composed of both linear and branched polymers of poly(ADP-ribose) (PAR). However, the biochemical mechanism of polymerization and biological functions of branched PAR chains are elusive. Here we show that PARP2 is preferentially activated by PAR and subsequently catalyzes branched PAR chain synthesis. Notably, the direct binding to PAR by the N-terminus of PARP2 promotes the enzymatic activity of PARP2 toward the branched PAR chain synthesis. Moreover, the PBZ domain of APLF recognizes the branched PAR chain and regulates chromatin remodeling to DNA damage response. This unique feature of PAR-dependent PARP2 activation and subsequent PARylation mediates the participation of PARP2 in DNA damage repair. Thus, our results reveal an important molecular mechanism of branched PAR synthesis and a key biological function of branched PARylation.

[1] Department of Cancer Genetics and Epigenetics, Beckman Research Institute, City of Hope Medical Center, Duarte, CA 91010, USA. [2] UMR7242, Biotechnology and Cell Signaling, École Supérieure de Biotechnologie de Strasbourg, CNRS/Strasbourg University, BP10413, 67412 Illkirch, France. Correspondence and requests for materials should be addressed to X.Y. (email: xyu@coh.org)

Poly(ADP-ribose) polymerases (PARPs) catalyze poly(ADP-ribosyl)ation of target proteins (PARylation) using NAD$^+$ as the donor of ADP-ribose[1,2]. ADP-ribose moieties are covalently linked to the side chains of several amino acid residues, such as Aspartic acid, Glutamic acid, Arginine, Lysine, and Serine[3–5]. Four members of the PARP family, including PARP1, 2, 5A, and 5B, are able to add additional ADP-ribose to the first ADP-ribose through 1, 2-glycosidic bond, and keep elongating the chain with up to one hundred ADP-ribose units; whereas, the rest of PARP family enzymes catalyze only mono(ADP-ribosyl) ation (MARylation)[6–8].

Although, the basal level of ADP-ribosylation is relatively low, PARPs can consume up to 90% of cellular NAD$^+$ upon DNA damage, synthesizing massive ADP-ribosylation, especially PARylation[9–11]. Since each ADP-ribose reside contains two negatively charged phosphate groups, poly(ADP-ribose) (PAR) chains add huge amounts of negative charge to DNA lesions. Electrostatic repulsion between negatively charged DNA and PAR leads to relaxation of chromatin structure[12–14]. Moreover, PAR is recognized by PAR-binding modules found in many chromatin remodeling complexes and DNA damage repair factors, which mediates the recruitments of these DNA damage repair machineries to the sites of DNA damage further facilitating chromatin remodeling and DNA damage repair[12,14–17]. Hence, PARylation plays a pivotal role in DNA damage repair[18,19].

Among the four PARPs known to catalyze PARylation, only PARP1 and PARP2 are localized in the nucleus and both of them participate in the early DNA damage response. Once DNA damage occurs, PARP1 is immediately recruited to the damaged DNA ends through its N-terminal DNA-binding motif, and stimulates PARylation via its C-terminal enzymatic domain[20,21]. The quick PARylation signal also mediates the recruitment of PARP2 to the damaged site and both of them act together to expand the PARylation signal at DNA lesions for the ultimate repair[22]. Like PARP1, PARP2 also contains an N-terminal regulatory domain and a C-terminal enzymatic domain[23]; however, the mechanism of activation of PARP2 and the role of PARP2-mediated PAR chain formation in DNA damage remains elusive.

Interestingly, PAR chains synthesized during the DNA damage repair (DDR) are not homogenously linear polymer forms of ADP-ribose (ADPr)[9,24]. The ADPr unit in the PAR chains has two ribose sugars. Each ribose in the unit is linked to the ribose sugar of an adjacent ADP-ribose (ribose–ribose glycosidic bond) forming the linear polymer. The distal ribose in one ADPr unit can also be linked with the distal ribose of another ADPr resulting in the branching of PAR[25,26]. However, how the branched PAR chain is synthesized and the biological function of the branched PAR remains unclear.

Here we examined the activation of PARP2, and found that PARP2 could be activated by PAR and subsequently catalyze the branched PAR chain formation. Moreover, the branched PAR is recognized by the PBZ domain of APLF for the removal of histone barrier during DNA damage repair.

## Results

**Loss-of-PARP2 affects the branched PAR chain formation**. To study the molecular mechanism of PARP2-mediated PAR polymerization, we thoroughly examined PARP2-dependent PARylation induced by hydrogen peroxide ($H_2O_2$) in wild-type mouse embryonic fibroblasts (MEFs) versus Parp2$^{-/-}$ MEFs using a panel of complementary approaches aimed to decipher the biochemistry of PAR (Fig. 1a). Compared to wide-type MEFs, we only observed a slight reduction of PARylation in the Parp2$^{-/-}$ MEFs using an anti-PAR antibody (monoclonal antibody, Trevigen 4335-MC-100) in dot blotting (Fig. 1b). In contrast, loss of

PARP1 abolished majorities of DNA damage-induced PARylation (Supplementary Figure 1). The results are in agreement with previous studies and have shown that PARP2 only contributes a set of PARylation events during DNA damage repair[27,28]. We also confirmed that the total PAR was slightly reduced in the absence of PARP2 using UV spectroscopy, whereas the PARP1 deficiency caused ~85% of total PARylation reduction (Fig. 1c).

To examine whether PARP2 mediates any specific PARylation events, we analyzed these samples with LC–MS/MS. Prior to mass spectrometry, the extracted PAR chains from the wild-type MEFs or the Parp2$^{-/-}$ MEFs were treated with venom pyrophophatase (PPase) and alkaline phosphatase (AP) to generate adenosine (Ado), ribosyladenosine (R-Ado) and diribosyladenosine (R$_2$-Ado), which represent the terminal, linear, or branched unit in PAR chain, respectively (Supplementary Figure 2A and 2B). Consistent with the results of dot blotting, we detected only a slight reduction of R-Ado in the Parp2$^{-/-}$ MEFs, suggesting that total PARylation is modestly impaired in the absence of PARP2. In contrast, the levels of Ado and R$_2$-Ado were significantly reduced (Fig. 1d), suggesting that the branched PAR chain polymerization is notably affected in the absence of PARP2. To further validate these results, we used the CRISPR-Cas9 system to knockout PARP2 in U2OS cells. Consistently, we found that loss of PARP2 impaired branched PAR chain formation in these cells as shown by the important decreased production of R$_2$-Ado (Fig. 1e). In addition, we reconstituted PARP2-deficient cells with full length PARP2 or the E545-to-A mutant (E545A), which disrupts the catalytic cage[23]. Only full length PARP2 but not the E545A mutant rescued the branched PAR chain formation (Fig. 1e). Collectively, these results indicate that PARP2 is absolutely needed for the formation of branched PAR chain.

Moreover, we quantitatively measured the branched chain (R$_2$-Ado) in both wide type and PARP2-null cells with LC–MS/MS. Based on a previously demonstrated approach[29], we compared the ratio between R$_2$-Ado and R-Ado, and found that the branched unit is ~2% of the linear chain unit in the wild-type cells. In PARP2-null cells, the branched site is <1% of the linear chain unit (Supplementary Figure 3).

**PARP2 is activated by PAR**. Having established the role of PARP2 in branched PARylation, we sought to characterize the molecular mechanism involved in PARP2 activation during this phenomenon. Different from PARP1, PARP2 has a short N-terminal motif and a WGR domain that may recognize nucleic acid. It has been shown that both 5′ phosphorylated DNA oligo (5′P-ssDNA) and ssRNA are able to activate PARP2[27,28]. Using the in vitro PARylation assay, we confirmed that either 5′P-ssDNA or ssRNA can trigger PARP2 activity (Supplementary Figure 4A). Similar to DNA and RNA, PAR is a type of nucleic acid polymer. Strikingly, we observed that PAR chain is sufficient to activate PARP2, but not PARP1, in the in vitro PARylation assay (Supplementary Figure 4A and 4B). Compared to the similar length of 5′P-ssDNA or ssRNA, PAR was more potent to activate PARP2 in the colorimetric assays (Supplementary Figure 4C). However, the auto-PARylation of PARP2 could not be induced by single ADPr (Supplementary Figure 4D). Moreover, in contrast to PARP2, the E545A mutant of PARP2 cannot be activated by PAR (Supplementary Figure 4E). This robust and preferential activation of PARP2 by PAR suggests that PARylation itself could be an important source to activate PARP2.

**PARP2 mediates the branched PAR chain formation**. As PAR is able to activate PARP2, we next investigated the identity of PARP2 substrate and hypothesized that the most favorite target

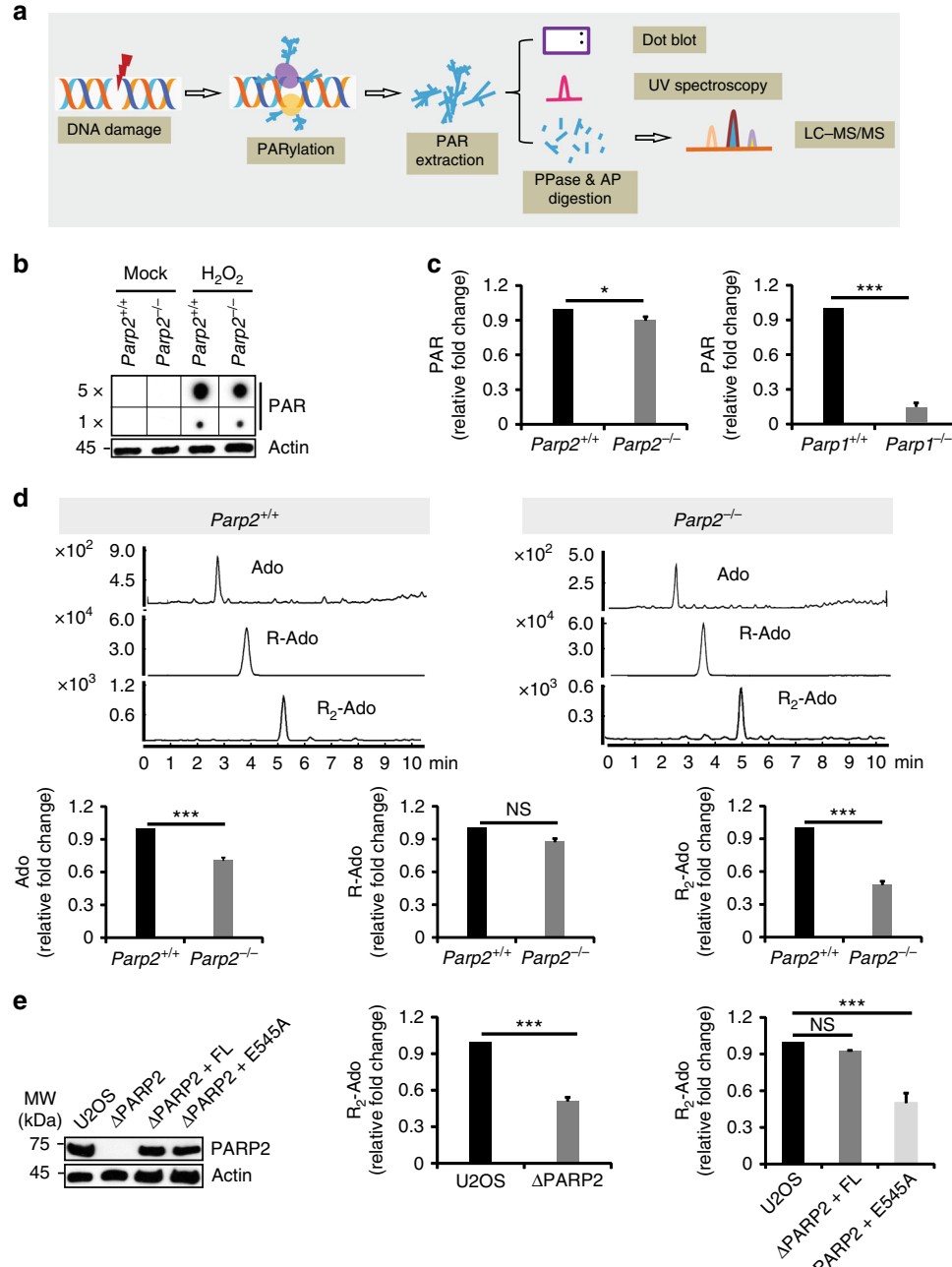

**Fig. 1** Loss-of-PARP2 affects the branched PAR chain formation. **a** Diagram depicting the procedure of sample preparation for dot blot, UV spectroscopy or mass spectrometry. Wide type or Parp2$^{-/-}$ MEFs were treated with 500 μM $H_2O_2$ for 10 min to induce DNA damage. PAR was extracted and followed by digestion with pyrophosphatase (PPase) and alkaline phosphatase (AP) prior to LC–MS/MS. **b** Dot blot assays were performed with anti-PAR antibody. 5X means five-fold loading samples. Actin was used as the control of cell lysates (monoclonal antibody, Sigma A2228). **c** Depletion of PARP2-induced minor reduction of total PAR in response to DNA damage, whereas lacking PARP1 remarkably suppressed total PAR level. The levels of PAR were examined at 259 nm using UV spectroscopy. **d** Mass spectrometry detection of adenosine (Ado), ribosyladenosine (R-Ado) and diribosyladenosine (R$_2$-Ado). LC–MS/MS quantitative analysis showed that Ado (terminal PAR unit) and R$_2$-Ado (branched PAR unit) were remarkably reduced in PARP2-deficient cells, but not R-Ado (linear PAR unit). **e** Only full length PARP2, but not the E545A mutant restores the branched PAR chain formation. PARP2 was deleted in U2OS cells and reconstituted with either full length PARP2 (ΔPARP2 + FL) or the E545A mutant (ΔPARP2 + E545A, left panel). The levels of R$_2$-Ado were measured by LC–MS/MS (middle and right panels). Data are represented as mean ± s.d. as indicated from three independent experiments. Significance of differences was evaluated by Student's $t$-test. NS: non significant; *statistically significant ($p < 0.05$); ***statistically significant ($p < 0.001$)

would be PAR itself. To test this possibility, we first excluded PARP2 after in vitro PARylation assay, harvested the activator PAR chains, and subjected them to the mass spectrometry (Fig. 2a). Interestingly, we found that the branched chains measured by the production of R$_2$-Ado were remarkably increased in

the activator PAR chains after incubation with PARP2 (Fig. 2b). In contrast, under similar assay condition, PARP1 alone could not add branched PAR chains on the existed PAR chains (Fig. 2c). Notably, the catalytic inactive mutant of PARP2 also failed to synthesize the branched chains (Fig. 2d). Taken together,

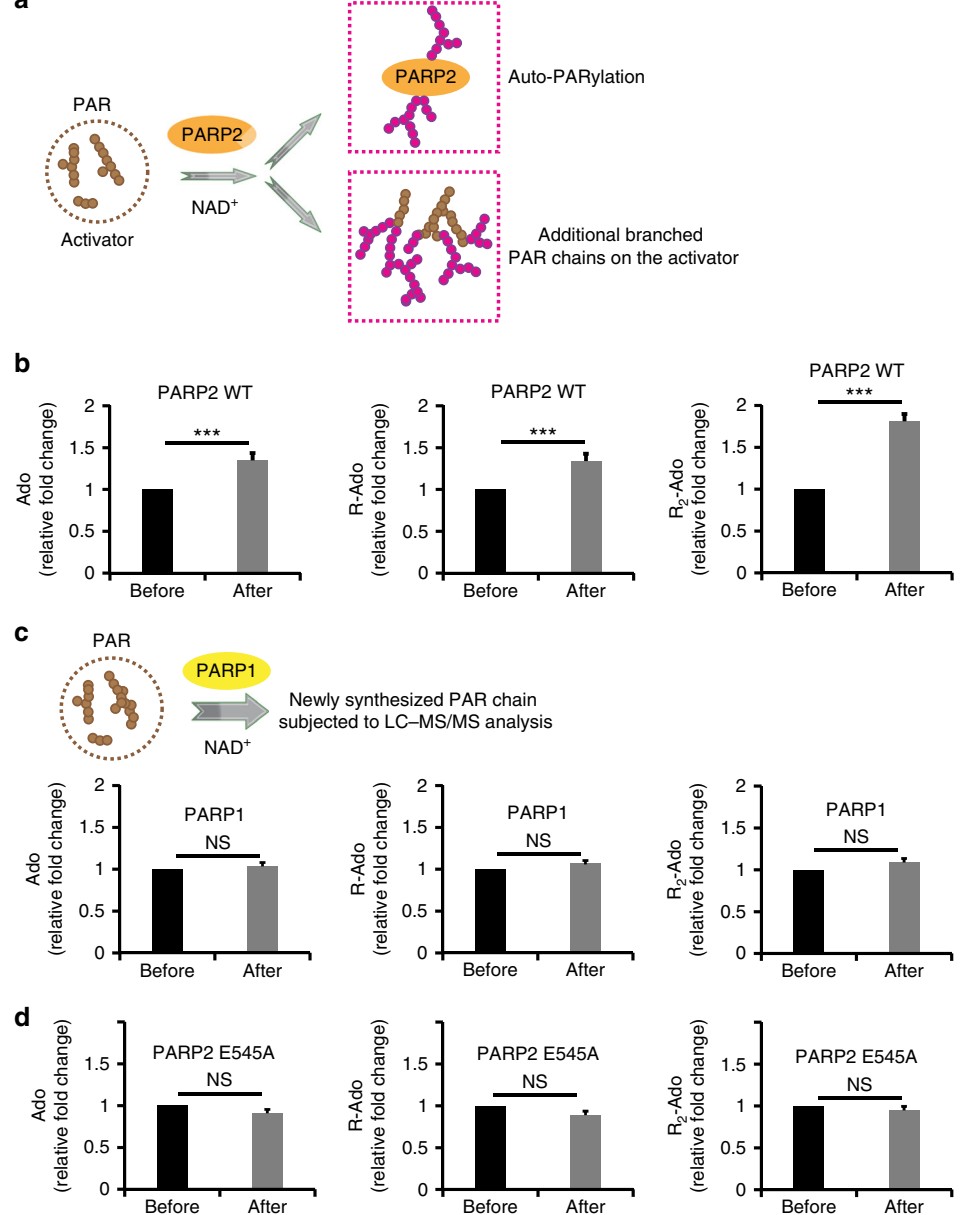

**Fig. 2** PARP2 mediates the branched PAR chain formation. **a** A diagram of PARP2-dependent branched PAR chain formation. **b** PARP2 catalyzes branched PAR chain polymerization. The activator PAR was examined by mass spectrometry following the in vitro PARP2-dependent PARylation. **c** PARP1 cannot PARylate the existed PAR chains in the same tested condition as **b**. **d** The PARP2E545A mutant fails to catalyze additional PAR chains on the activator PAR. Data are represented as mean ± s.d. as indicated from three independent experiments. Significance of differences was evaluated by Student's $t$-test. NS: non significant; ***statistically significant ($p < 0.001$)

these results suggest that PARP2 is able to add additional branched chains on top of the activator PAR. Due to the polymerization of branched chains, additional Ado and R-Ado were also increased in the activator PAR (Fig. 2b).

In addition, we also measured the branching sites in PARP1-dependent PARylation. We measured the ratio between $R_2$-Ado and R-Ado the in vitro PARylation assays. With only PARP1, we found that the branched site is ~1% of the linear unit when only PARP1 was used to catalyze the PARylation (Supplementary Figure 5A). When we added PARP2, we found that branched site increased ~2-fold (Fig. 2b). Moreover, we over-expressed PARP1 in the PARP2 KO cells, and found that the branched sites were increase with additional PARylation catalyzed by PARP1 (Supplementary Figure 5B). The results suggest that PARP1 is

also able to catalyze branched chain, but at lower rate[30]. Moreover, other PARPs, such as PARP3 and PARP10, also participate in DNA damage repair[31–33]. It is possible that other PARPs catalyze single or oligo branch units on top of the existed PAR chains.

**Branched PARylation is regulated by the N-terminus of PARP2.** Besides the C-terminal catalytic domain, PARP2 has a short N-terminal region (NTR) and a WGR domain. Similar to the Zinc Finger motifs of PARP1, the NTR and the WGR domains of PARP2 recognize PARP2 activators and induce conformational changes in the catalytic domain for the activation of PARP2-dependent PARylation[23,28]. To study if the NTR and

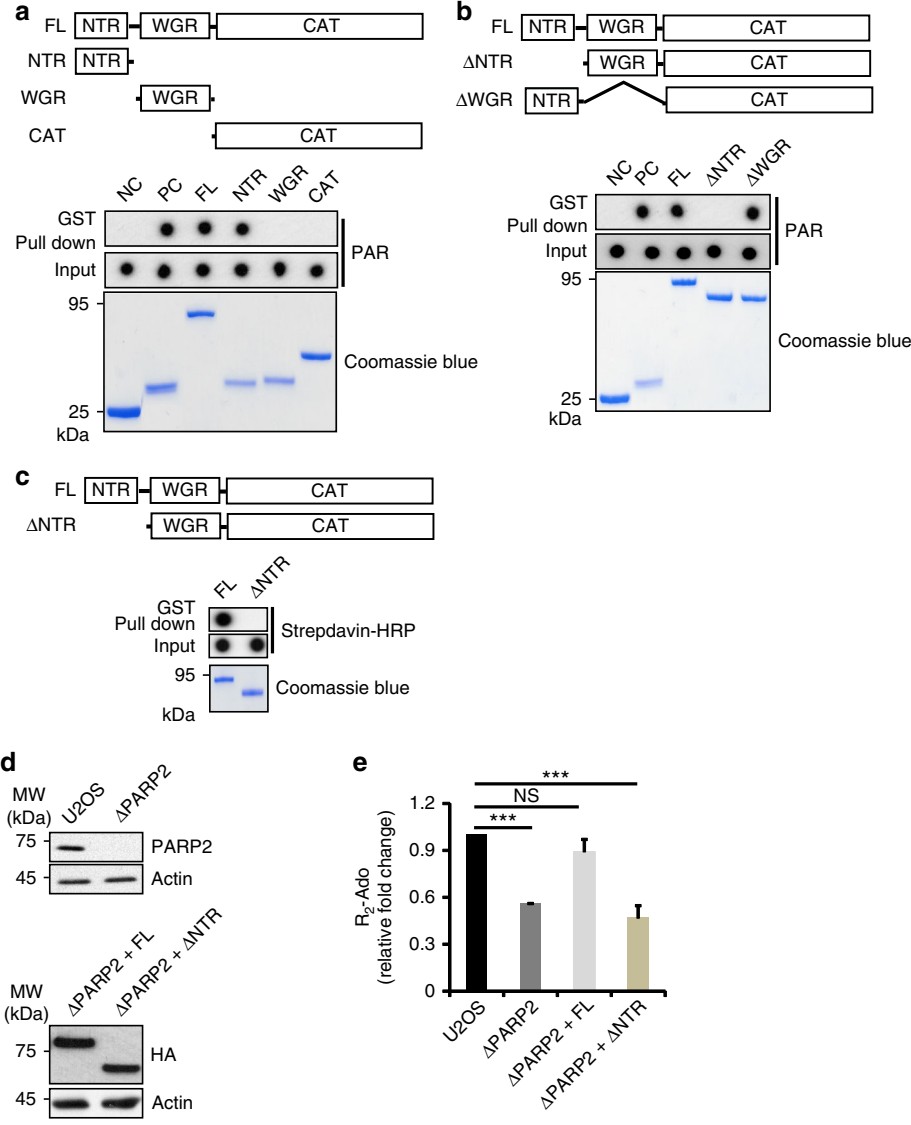

**Fig. 3** N-terminus of PARP2 mediates the branched PAR chain formation. **a** The NTR domain of PARP2 recognizes PAR. The recombinant GST fusion proteins were incubated with $^{32}$P labeled PAR. Protein-PAR complex was pulled down by glutathione agarose beads followed by autoradiography (top panel). Recombinant GST and GST-RNF146 WWE domain were used as the negative and positive controls (NC and PC), respectively. The GST fusion proteins were also examined by the SDS-PAGE followed with Coomassie blue staining (bottom panel). **b** The NTR region of PARP2 is required for the interaction with PAR. Full length PARP2, $\Delta$ NTR or $\Delta$ WGR were incubated with $^{32}$P labeled PAR. Associated PAR was examined (top panel). Recombinant PARP2 was stained with Coomassie blue (bottom panel). **c** The NTR domain is required for the activation of PARP2. Biotin-NAD$^+$ was used in the in vitro PARP2-dependent PARylation assays. PARP2-dependent PARylation was examined by dot blot using streptavidin-HRP. The proteins were also examined by the SDS-PAGE followed with Coomassie blue staining (bottom panel). **d** The levels of PARP2 in U2OS and PARP2-null cells were examined by western blot (upper panel). The PARP2-null cells were reconstituted with full length PARP2 or the $\Delta$ NTR mutant. And their expression was confirmed by western blot (lower panel). **e** The NTR domain is required for the branched PAR chain synthesis in response to DNA damage. Cells were treated with 500 μM H$_2$O$_2$ for 10 min. PAR was extracted and processed by PPase and AP. R$_2$-Ado was quantified by LC–MS/MS examination. Data are represented as mean ± s.d. as indicated from three independent experiments. Significance of differences was evaluated by Student's $t$-test. NS: non significant; ***statistically significant ($p < 0.001$)

the WGR domain mediate PAR-dependent activation of PARP2, we examined the interaction between PAR and these two domains, and found that the NTR interacted with PAR (Fig. 3a). Deletion of the NTR but not the WGR domain abolished the interaction between PAR and PARP2, suggesting that the NTR of PARP2 is required for the interaction of PARP2 with PAR (Fig. 3b).

Next, we examined whether the NTR played a role for the PARP2-dependent PARylation. The recombinant full length PARP2 or the NTR deletion mutant were generated and

examined in the in vitro PARylation assays. Consistent with the PAR-binding results, lacking the NTR domain abolished PARP2-dependent PARylation (Fig. 3c).

We also examined the functional requirement of the NTR in the PARP2-dependent PARylation in cells. Full length PARP2 or NTR deletion mutant were reintroduced into PARP2-deficient cells and the expression of the proteins were verified by western blot (Fig. 3d). We found that the NTR deletion mutant could not rescue the PARP2-dependent PARylation in these cells, including the PARP2-dependent branched chain formation (Fig. 3e).

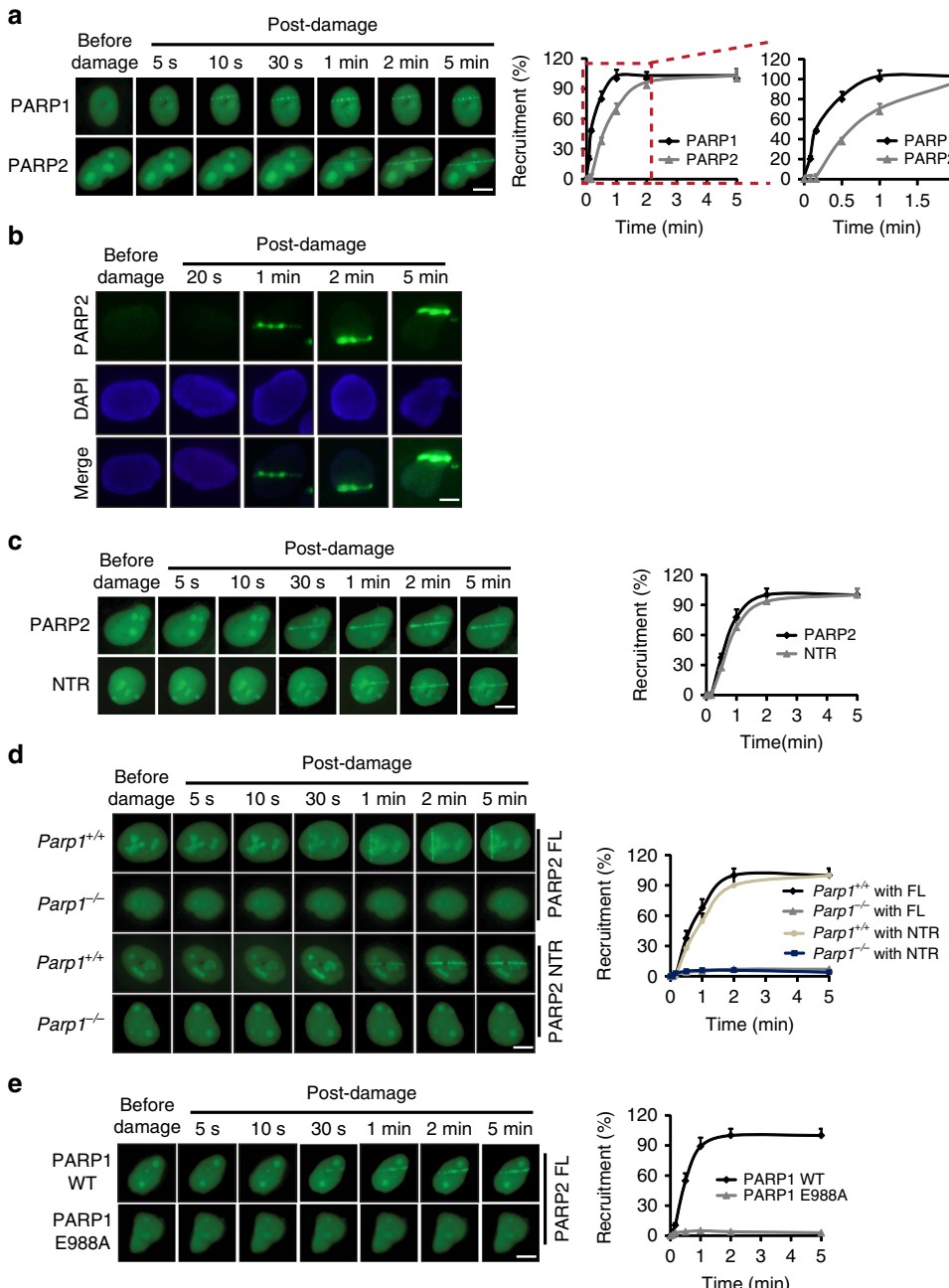

**Fig. 4** The recruitment of PARP2 to DNA lesions is mediated by PARP1-dependent PARylation. **a** The recruitment kinetics of PARP2 to DNA lesions. GFP-PARP1 or PARP2 were expressed in U2OS cells. Following laser micro-irradiation treatment, the recruitment of PARP1 or PARP2 was examined with live-cell imaging at the indicated time points (left panel). The relocation kinetics is shown in the right panel. Data represent mean ± s.d. from three biologically independent experiments (right panel). At least 20 cells were included in each experiment. **b** The recruitment of endogenous PARP2 was examined following laser micro-irradiation by anti-PARP2 antibodies. **c** The NTR domain alone is sufficient to be recruited to the DNA damage sites. The recruitment kinetics of GFP-NTR at different time points was measured. **d** PARP1 is required for the recruitment of PARP2. GFP-PARP2 or the NTR domain was expressed in the $Parp1^{+/+}$ or $Parp1^{-/-}$ MEFs. The recruitment kinetics was examined at indicated time points. The percentage of recruitment under different PARP1 conditions is shown in the right panel **e**. The catalytic activity of PARP1 is required for the recruitment of PARP2. PARP1-deficient U2OS cells were stably expressing the full length PARP1 or the E988A mutant (catalytic inactive mutant). GFP-PARP2 was expressed in the cells, and the recruitment kinetics was measured. Scale bar represents 5 μm

Collectively, the NTR is required for the PARP2-dependent PARylation, especially the branched chain formation.

**PARP1-dependent PARylation mediates the recruitment of PARP2 to the sites of DNA damage.** Since, PARP2 recognizes PAR, we asked whether PARylation facilitates the recruitment of

PARP2 to the sites of DNA damage. It has been reported that PARP1 reaches the sites of DNA damage within one second following laser micro-irradiation[34]. With the laser micro-irradiation, we examined the kinetics of PARP2 recruitment at the damage site, and found that unlike PARP1, PARP2 started to accumulate at around 30 s and reached the peak level at around 2 min (Fig. 4a). Similar recruitment kinetics of endogenous PARP2

was observed (Fig. 4b). Moreover, the NTR domain alone was sufficient to be recruited to the damaged sites (Fig. 4c). And the recruitment of either full length PARP2 or NTR was disrupted with the treatment of PARP inhibitor olaparib (Supplementary Figure 6A). Collectively, these results suggest that the NTR of PARP2 may recognize locally produced PAR for the recruitment of PARP2.

Compared to PARP2, we observed that PARP1 is recruited to the sites of DNA damage much earlier (Fig. 4a). At around 20 s following DNA damage, we were able to clearly detect PARylation at lesions (Supplementary Figure 6B), while PARP2 had not been recruited at the same time point. It indicates that PARP1 mediates the early PARylation at lesions. As PARylation may mediate the recruitment of PARP2, we next examined the recruitment kinetics of PARP2 in PARP1-deficient cells. Both the full length and the NTR of PARP2 were recruited to the sites of damage in the presence of PARP1. However, loss of PARP1 largely impaired the recruitment of PARP2 (Fig. 4d), indicating that the recruitment of PARP2 is dependent on PARP1.

To validate our hypothesis, we re-expressed wild type or the catalytically inactive mutant PARP1 (E988A) in PARP1-deficient cells, and found that only wild type but not the PARP1E988A mutant can rescue the recruitment of PARP2 (Fig. 4e), suggesting that the catalytic activity of PARP1 is required for the recruitment of PARP2. Of note, the recruitment kinetics of PARP1 was not affected by the E988A mutation (Supplementary Figure 6C). Taken together, these results confirm that the NTR of PARP2 recognizes the PARP1-dependent PARylation, which mediates the recruitment of PARP2 to the sites of DNA damage.

**APLF recognizes the branched PAR chain**. Although, the branched PAR chain has been identified more than 35 years ago[26], the biological function and the binding module of the branched PAR chain are still unclear. One unique feature of chemical structure in the branched chain is that three ADPr units are linked to each other. Based on the structural analysis, until now all the PAR-binding modules have been reported to recognize only one or two ADPr units[15,35–41]. These observations open up the possibility that multiple PAR-binding modules may be coordinating together to interact with the branched site of the PAR chain. Structural studies have demonstrated that APLF contains two tandem PBZ motifs, in which the first PBZ motif binds to two ADPr units, and the second one recognizes the third ADPr[36]. Based on these studies, we hypothesize that the tandem PBZ motifs of APLF may be the readers of branched sites in a PAR polymer.

To test the hypothesis, we generated recombinant APLF PBZ protein and incubated it with branched PAR chains. The APLF PBZ-PAR complex was harvested and partially digested with PARG. The APLF binding PAR residues were eluted and digested with PPase and AP, and analyzed by LC–MS/MS (Fig. 5a). Interestingly, the $R_2$-Ado but not Ado or R-Ado, was remarkably enriched compared with the controls (Fig. 5b), indicating that APLF PBZ preferentially binds the branched sites over linear PAR chains. The specificity of this binding was confirmed by using the WWE domain of RNF146 as a control that is known to recognize only iso-ADPr; the linker between tandem ADPr in the PAR chains[37]. In agreement with previous studies, only R-Ado was isolated by the pull down with the WWE domain under the same assay condition (Fig. 5c).

To further analyze the tandem PBZ motifs of APLF, we mutated the key residues in each PBZ motif, and examined the PAR-binding activities. We found that loss of the first PBZ motif (PBZ1) remarkably reduced the total PAR-binding, whereas loss of the second PBZ (PBZ2) also impaired the interaction with PAR

(Supplementary Figure 7A). With mass spectrometry, we found that loss of either PBZ motif abolished the recognition of branched sites (Supplementary Figure 7B). Notably, loss of PBZ2 had slightly increased association with Ado and R-Ado (Supplementary Figure 7B). It indicates that loss of one PBZ, especially PBZ2, may switch the recognitions of APLF from the branched sites to linear chain or terminal ADPr.

APLF acts as a histone chaperon and nuclease during DNA damage repair[42]. Previous study showed that PARylation mediates the recruitment of APLF to the sites of DNA damage[18,43]. Since, our data suggest that PARP2 mediates the branched PAR chain formation, and APLF recognizes branched PAR chains; we examined whether PARP2 mediates the recruitment of APLF to the sites of DNA damage. We examined the recruitment kinetics of APLF in PARP2-deficient cells, and found that lacking PARP2 clearly impaired the recruitment of APLF to the sites of DNA damage (Fig. 5d), which is also in agreement with previous studies[44]. Collectively, the results suggest that the branched PAR chains mediate the recruitment of APLF to the sites of DNA damage.

**PARP2-dependent branched PAR chain polymerization is important for histone H3 removal in response to DNA damage**. APLF is a DNA damage repair specific histone chaperone that preferentially binds to the H3/H4 tetramer, and mediates histone barrier eviction for relaxing the chromatin structure[45]. Since our data suggest that the recruitment of APLF relies on the branched PAR chains, we examined whether PARP2-dependent branched PAR chains mediate histone eviction at the sites of DNA damage. To induce DNA damage, we used doxycycline inducing system to express Cas9 nuclease in PARP2-deficient or proficient cells. With the specific gRNA, the controlled Cas9 nuclease generated a DNA double-strand break (DSB) at the unique AAVS1 locus on the chromosome 19 (Fig. 6a). The solitary DSB was confirmed by the phosphorylation of $H_2AX$ at the flanking regions (Fig. 6b). This system would remove the nucleosomes within 300 bp at each side of the DSB[46]. Interestingly, histone H3 removal was largely suppressed in PARP2-deficient cells (Fig. 6c), suggesting that this histone H3 removal event was dependent on PARP2. Consistently, loss of APLF also suppressed histone H3 removal at the sites of DNA damage (Fig. 6d). In contrast, without DNA damage, histone H3 was present at this locus, and lacking PARP2 or APLF did not affect H3 deposition when DNA damage did not occur (Supplementary Figure 8A).

Because PARylation is a very transient posttranslational modification, we could not directly detect APLF at the sites of DNA damage in this assay system. However, these results convincingly suggest that PARP2 mediates branched PAR chain formation may regulate the APLF-dependent histone H3 removal at the sites of DNA damage. Moreover, we used q-PCR with primers on each side of the DSB to quantitatively measure the repair kinetics (Fig. 6e), and found that the loss of either PARP2 or APLF significantly delayed the DNA damage repair (Fig. 6f). To exclude any off-target effect, we reintroduced full length PARP2 or APLF in the deficient cells. And full length PARP2 or APLF were able to rescue the DNA damage repair (Supplementary Figure 8B). In addition, the tandem PBZ motifs of APLF mediated this molecular event as well (Supplementary Figure 8C).

To further validate the DSB repair function of PARP2 and APLF, we treated APLF and PARP2-deficient cells with IR and used neutral comet assays to examine the DSB repair. Again, we found that the DSB repair kinetics in APLF and PARP2-deficient cells were similar (Supplementary Figure 9A). The results are consistent with previous publications that both APLF and PARP2

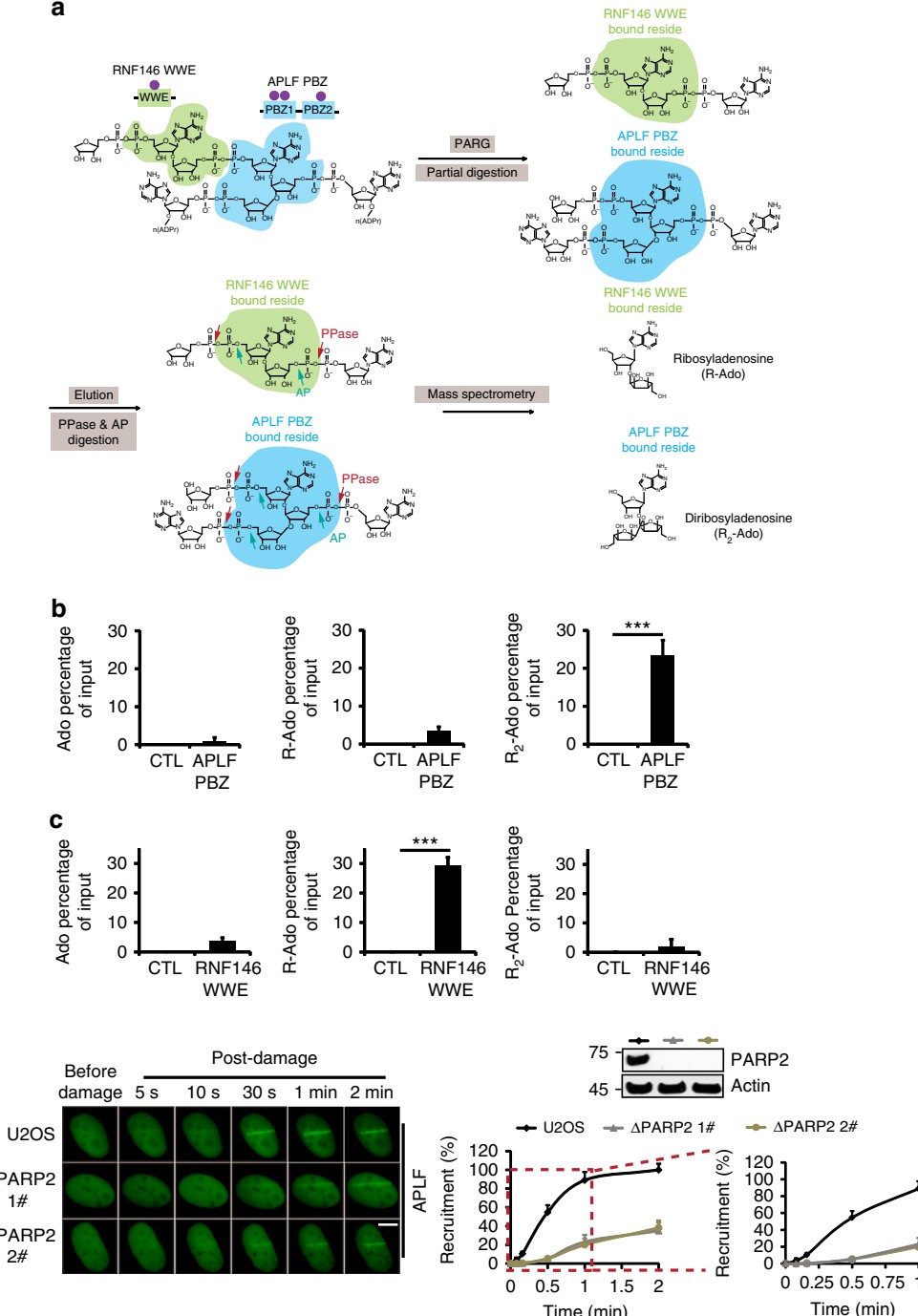

**Fig. 5** APLF recognizes the branched PAR. **a** Diagrammatic representation of the sample preparation. PAR was incubated with APLF PBZ or the WWE domain of RNF146, and then partially digested by PARG. The protein-associated residues were further treated with PPase and AP, and then subjected to LC–MS/MS. APLF PBZ (**b**) or the WWE domain of RNF146 (**c**) bound residues were examined. **d** The recruitment of APLF to DNA lesions is dependent on PARP2. GFP-APLF was expressed in U2OS or PARP2-null cells. The recruitment kinetics of APLF was examined following DNA damage at various time intervals. Scale bar represents 5 μm. Data are represented as mean ± s.d. as indicated from three independent experiments. Significance of differences was evaluated by Student's $t$-test. NS: non significant; ***statistically significant ($p < 0.001$)

are important for DSB repair[47–49]. To further examine whether the PARP2-APLF pathway is involved in HR and NHEJ, we depleted PARP2 or APLF in DR-GFP U2OS and EJ5-GFP U2OS cells, respectively. Based on these GFP reporter assays, we found that NHEJ was clearly impaired when cells lost PARP2 or APLF. However, HR was also mildly suppressed when cells were lacking PARP2 or APLF (Supplementary Figure 9B). The results are in agreement with previous studies on PARP2 and APLF[49–52].

Taken together, these results suggest that PARP2-dependent branched PAR chains play a potentially important role in DNA damage repair.

## Discussion

The major finding of this study is that PARP2 is involved in branched PAR synthesis which in turn governs APLF mediated

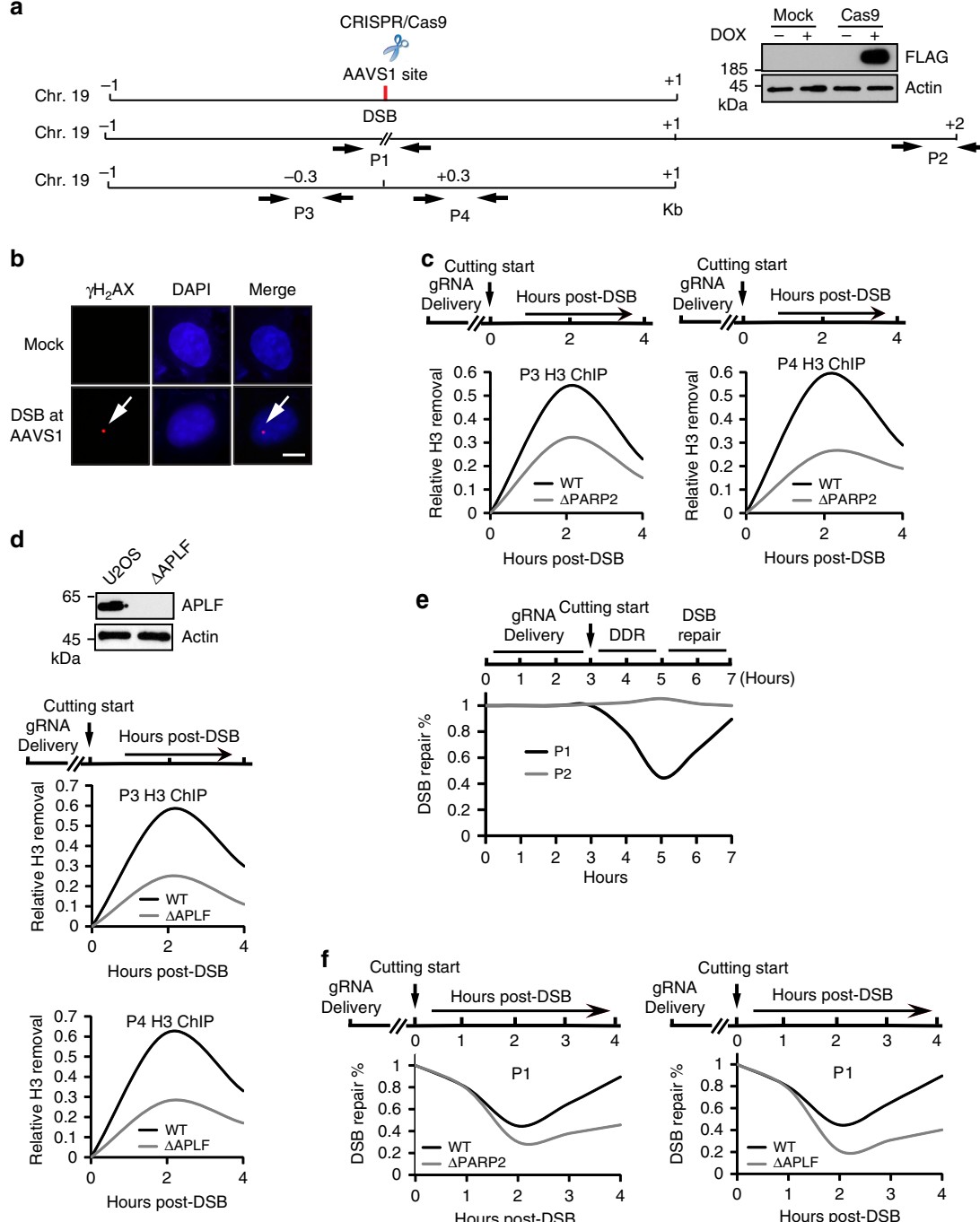

**Fig. 6** PARP2-dependent branched PAR chain is important for histone H3 removal in response to DNA damage. **a** Doxycycline (DOX)-inducible Cas9 (iCas9) was expressed in the U2OS or U2OS PARP2-null cells. The expression of iCas9 was examined by western blot using anti-FLAG antibody (monoclonal antibody, Sigma F1804, the inset). A gRNA facilitates a single DSB as the AAVS1 locus. P1 represents a pair of primers flanking both sides of the DSB for the analysis of DSB repair. P2 is a pair of primers specific to a region 2 kb upstream to the DSB and serves as a negative control of DSB repair. P3 and P4 are two sets of primers located at 0.3 kb upstream and downstream respectively from the DSB. These two primers were used for the detection of histone H3 removal in response to DNA damage. **b** γH$_2$AX (polyclonal antibody, Abcam ab11174) was examined as a surrogate maker of the solo DSB at AAVS1 locus using immunofluorescence analysis. **c** Histone H3 removal is severely compromised in the absence of PARP2. Histone H3 removal was examined by ChIP assays with antibody (polyclonal antibody, Millipore 06-755) using P3 and P4 primers. **d** Depletion of APLF also suppressed histone H3 removal at the DSB. Depletion of APLF was verified by western blot using anti-APLF antibody (polyclonal antibody, Thermo Fisher Scientific PA5-39776, left panel). **e** DSB repair was monitored by q-PCR using P1 and P2. **f** Loss of either PARP2 or APLF significantly suppresses DSB repair. DSB repair in cells lacking PARP2 or APLF was examined by q-PCR. Scale bar represents 5 μm

histone H3 removal during DNA damage repair. In particular, PARP2 itself gets activated by PAR and plays an important role in the branched PAR chain polymerization. Previous studies have reported that PARP2 can be activated by nucleic acids, including DNA and RNA. Here we demonstrate that PAR can act like other nucleic acids and activate PARP2. We also provide evidence that the NTR domain of PARP2 is able to recognize PAR and facilitates the activation of PARP2. In fact, ADPr itself is a unique type of nucleic acid, and it is likely that the NTR utilizes the common binding pocket to interact with PAR, DNA, or RNA. Additionally, the NTR domain is required for the localization of PARP2 to the sites of DNA damage. This process is also largely dependent on PARP1-mediated PARylation. Taken together, it is likely that the NTR of PARP2 recognizes PARP1-mediated PARylation following DNA damage, which subsequently targets PARP2 to the sites of DNA damage and facilitates the activation of PARP2 (Supplementary Figure 10).

Once PARP2 is activated by PAR, it catalyzes additional PARylation on top of the existing PAR chains, resulting in formation of branched PAR chain. With mass spectrometry analysis, we have shown that PARP2 is able to catalyze branched PAR formation in the in vitro PARylation assays. Moreover, lacking PARP2 in cells suppresses the branched PAR chain formation. Thus, it is likely that PARP2 acts as the secondary PARP during PARylation and synthesizes additional PAR chains (Supplementary Figure 10). Previous studies have demonstrated de novo PARylation in absence of PARP1, albeit at much lower level[10]. Thus, there is a possibility that PARP2 recognizes DNA or RNA to catalyze de novo PARylation in the absence of PARP1. Besides PARP2, other PARPs including PARP7, 9, 11, 12, 13, 14, and 15 may also act as the secondary PARPs. These PARPs contain the Macro domain and/or the WWE domain both of which are PAR-binding modules[5,6,16,24]. Besides PARP1 and PARP2, PARP3, and PARP10 have been shown to participate in DNA damage repair. It is also possible that these PARPs may recognize PARylation and catalyze additional ADP-ribosylation on top of the PAR chains.

Although PARP1 is also able to catalyze the branched chain, we found that the level of branched chain was relatively low in PARP2-deficient cells. The structures of the catalytic pockets of PARP1 and PARP2 have been examined. In fact, the detailed structures are quite different in PARP1 and PARP2. The catalytic pocket can be divided into acceptor site and donor site. Although, the binding sites of $NAD^+$ donor are quite similar, the substrate acceptor sites are quite different. Compared to that in PARP1, PARP2 has a unique extended loop with six additional residues (Leu523-Thr529) in the acceptor pocket that changes the tertiary structure[53]. It is very likely that the different orientation of substrate acceptor pocket determine the linear and branched PAR chain formation.

The branched site of PAR chain has a unique chemical structure with three ADPr clustered around, indicating that it may act as a signal and be specifically recognized by some motifs. Here we have shown that the tandem PBZ motifs of APLF recognize the branched PAR site. Previous studies have shown that the first PBZ motif of APLF recognizes two ADPr units; whereas, the second one only recognizes one ADPr unit because of a proline residue substation at the conserved ADPr-binding pocket[36]. Loss of either PBZ motif abolishes the binding with the branched sites. Interestingly, loss of PBZ2 does not abolish the recognition of PAR. Instead, it switches the PAR recognition from the branched site to the linear chain. Thus, it would be intriguing to elucidate the structure of this unique binding module in future. Besides the tandem PBZs motifs of APLF, other motifs may also recognize the branched PAR chains, especially those containing three ADPr-binding modules. An interesting example of this kind of

PARP is, PARP14 containing three tandem Marco domains, which may be involved in recognizing the branched sites. Uncovering other branched site-binding domains will further elucidate the biological function of the branched PAR chain.

Previous studies have shown that APLF is involved in histone eviction during DNA damage repair[43], our results and the results from other groups suggest that APLF may recognizes the PARP2-mediated PARylation during DNA damage repair[44]. Moreover, with a site specific system to generate a solitary DSB, we found that the PARP2-mediated PARylation regulates histone H3 removal at the sites of DNA damage. Thus, it is possible that APLF acts as a reader and downstream effector of the PARP2-mediated PARylation for histone H3 removal. Of note, loss of PARP2 arrests spermatogenesis during the last few steps[54,55], at which the majorities of nucleosomal histones have to be replaced by histone like proteins including transition protein 1 and 2 as well as protamine 1 and 2. These studies and our data are in strong support of a key contribution of PARP2 in histone removal. Although, the molecular mechanism of this chromatin remodeling event during spermatogenesis mimics the histone removal process at the sites of DNA damage, there are obvious differences in these events. While, the histone removal during spermatogenesis is a global event; the histone removal at the sites of DNA damage is a locally confined event. Our belief stems from the fact that similar phenomena of chromatin remodeling has been described for other DNA damage repair factors, such as RNF8, which mediates histone removal in both spermatogenesis and DNA damage repair[56,57].

Moreover, our results show that similar the repair defects have been observed in PARP2 or APLF-deficient cells. However, theoretically, DSB repair defects in PARP2-deficient cells should be milder than that in APLF-deficient cells, because lacking PARP2 only reduces >50% of branched chains, and a set of APLF was still able to be recruited to the sites of DNA damage. However, it is possible that besides APLF, other repair factors may also recognize the PARP2-dependent branched chain. Lacking PARP2-dependent branched chain may also impair the recruitments of other branch chain readers. In other words, APLF and PARP2 have overlapping but also independent functions in DNA damage repair. It is also possible that PARP2 ADP-ribosylates one substrate for the recruitment of APLF. But our results suggest that the PBZ motif of APLF specifically recognizes the branched chain formation (Fig. 5). Moreover, loss of the PBZ motif of APLF abolishes the function of APLF in DNA damage repair. Thus, it is likely that the activation of APLF relies on the interaction between the PBZ and the branched PAR chain. Further analysis on APLF-dependent pathway and identification of the PARP2 substrates will elucidate the detailed molecular mechanism.

## Methods

**Cell culture**. Mouse embryonic fibroblasts (MEFs) were derived from PARP1- or PARP2- wide-type or knockout mice. U2OS and 293T cells were purchased from ATCC and maintained in DMEM medium supplemented with 10% fetal bovine serum at 37 °C in 5% $CO_2$. To generate knockout cell lines, U2OS cells were transfected with PX459 vector containing either PARP2-sgRNA for the PARP2 knockout or APLF-sgRNA for APLF knockout. Transfected cells were plated at low density in 1.5 μg/ml puromycin. Single colonies were propagated, and individual clones were analyzed by western blotting. Depletion of PARP2 or APLF in U2OS cells was validated by western blotting using anti-PARP2 antibody (monoclonal antibody, Millipore MABE18) or anti-APLF antibody (polyclonal antibody, Thermo Fisher Scientific PA5-39776). U2OS PARP2- or APLF-null cells were transfected with vectors encoding HA-tagged either full length wide-type or mutants. Stable cell lines were established with 200 μg/ml hygromycin B selection. Western blotting was conducted to validate the efficiency of the reconstitution of indicated protein. To generate doxycycline-inducible cell lines, doxycycline was added to media to induce the Cas9 expression at a final concentration of 1 μg/ml for 24 h.

**Vector constructs**. To generate GFP-, HA-, GST-, and His-tagged proteins, DNA fragment containing the full-length PARP2 (NM_001042618.1) were cloned in-frame with tags into pEGFP-C1, pcDNA3.1/Hygro$^{(+)}$, pGEX-4T-1 and pET28a vectors, respectively. GST-tagged NTR (1–100 aa), WGR (100–220 aa) and CAT (201–570 aa) were cloned into pGEX-4T-1 vector. Full-length APLF, the PBZ domain of APLF (335–511 aa) and the WWE domain of RNF146 (98–190 aa) were inserted into pGEX-4T-1 vector to express recombination proteins. The NTR domain of PARP2 was cloned into pEGFP-C1 with nuclear localization sequence (PKKKRKV) at N terminus. Deletion mutants or point mutation of PARP2 or APLF were generated using site-directed mutagenesis kit. The corresponding primers are listed in Supplementary Table 1 and 2. pSpCas9(BB)-2A-Puro (PX459) V2.0 (plasmid #62988) and pCW-Cas9 (plasmid #50661) were purchased from Addgene. The sgRNA sequence for PARP2 knockout was 5′-CGACTATACCA TGACCTTGC-3′. The sgRNA sequence for AAVS1 knockout was 5′-GTTAATGTGGCTCTGGTTCT-3′. The sgRNA sequence for APLF knockout was 5′-TACCATTGAAGCCAAATCTA-3′. The siRNA sequence targeting PARP2 was 5′-GGAGAAGGAUGGGUGAGAAAdTdT-3′. The siRNA sequence targeting APLF was 5′-CAAUCAGUGGGAGGUAAUGUdTdT-3′. The corresponding antibodies are listed in Supplementary Table 4.

**Recombinant protein production**. All recombinant proteins were expressed in BL21 cells. His-tagged PARP1 or PARP2 was purified using Ni$^{2+}$-NTA chromatography. GST fusion proteins were purified using Glutathione Sepharose 4B. All recombinant proteins were examined by SDS-PAGE followed by Coomassie Blue staining.

**PAR synthesis and purification**. PAR synthesis reaction was carried out in a 2 ml mixture containing 100 mM Tris-HCl, pH 8.0, 150 mM NaCl, 10 mM MgCl$_2$, 2 mM NAD$^+$, 5 μM DTT, 50 μg octameric activator oligonucleotide GGAATTCC and 1 mg PARP1. The reaction was incubated at 30 °C for 40 min and stopped by addition of 2 ml ice-cold 20% TCA. Oligo DNA was cleaved by DNase I and proteins were digested by proteinase K. Following precipitation the pellet was washed with ice-cold pure ethanol. PAR was detached from histones or PARP1 protein using 0.5 M KOH/50 mM EDTA and extracted with phenol-chloroform-isoamyl alcohol. Purified PAR was fractionated according to chain length by anion exchange HPLC protocol[58]. 22-mer length PAR was chosen as activator in PARP2 ADP-ribosylation in vitro assay. The accurate length of PAR was confirmed by 20% native PAGE and the band was visualized using Pierce™ Color Silver Stain Kit (Thermo Fisher Cat#24597) according to the manufacturer's instructions.

**In vitro ADP-ribosylation assay**. The auto-PARylation assay was performed using 60 nM PARP2 or PARP2 mutant protein incubation with indicated activators (200 nM PAR, 5′P-ssDNA, ssRNA or ADPr) in PAR reaction buffer (100 mM Tris-HCl, pH 8.0, 150 mM NaCl, 10 mM MgCl$_2$, 500 μM DTT, and 0.125 μM $^{32}$P-NAD$^+$). The reaction was carried out for 20 min at 30 °C and stopped by the addition of SDS-loading buffer. The products were separated in SDS-PAGE gel and subjected to autoradiography. The protein in each reaction was stained by Coomassie blue.

To test the possibility of PARP2 or Δ NTR PARP2 could catalyze branched PAR chains formation on the activator PAR, biotinylated-NAD$^+$ was used as substrate in the reaction. The newly synthesized biotinyl-PAR was coupled with streptavidin beads for 1 h, and then was released by boiling in the SDS-loading buffer without bromophenol blue. The dot blotting was performed using streptavidin-HRP. The protein in each reaction was stained by Coomassie blue.

For PAR-dependent activation of PARP2 assay using LC–MS/MS detection, 200 nM PAR as activator and 100 nM PARP2, PARP2 E545A mutant or PARP1 were included in the reaction. The newly synthesized PAR on the existing PAR activator was examined using LC–MS/MS.

**PAR-binding assay**. Briefly, $^{32}$P-PAR was synthesized and purified using $^{32}$P-NAD$^+$ as substrate. Approximately 200 nM of each recombinant protein was incubated with 500 nM $^{32}$P-PAR, and 25 μl Glutathione agarose in the PBS buffer. The reaction mixture was incubated for 1 h followed by extensive washing of beads with PBS. The protein-PAR complex was released by heating at 90 °C for 5 min in the SDS-loading buffer without bromophenol blue. One microliter of sample was spotted onto the nitrocellulose membrane and subjected to autoradiography. The protein in each reaction was stained by Coomassie blue.

**Colorimetric PARP2 modification assay**. The catalytic activity of PARP2 measurement was carried out using a colorimetric assay which can detect the incorporation of biotinylated-NAD$^+$ into PAR. A total of 60 nM of His-tagged PARP2 protein was coupled on the Ni$^{2+}$-NTA-His 96-well plates (5′). The indicated activators (200 nM PAR, 5′P-ssDNA or ssRNA) and NAD$^+$ at various concentrations (15–1000 μM) were added in the 50 μl PAR reaction buffer. The ratio of NAD$^+$ to biotinylated-NAD$^+$ was 99:1. Reactions were processed[59]. The absorbance at 450 nm was recorded and fitted in the Michaelis-Menten model to yield $K_m$ and $V_{max}$ values. $V_{max}$ was divided by the molecular weight of PARP2 to calculate $K_{cat}$. The kinetic parameters presented represent the average of three independent experiments.

**Purification of intercellular PAR**. Cultured cells were stimulated with 500 μM H$_2$O$_2$ for 10 min, then the media were removed and cells were rinsed three times with PBS. Cells were collected by adding 20% TCA. Subsequently, the pellets were suspended in 0.5 M KOH/50 mM EDTA and incubation at room temperate for 2 h. A total of 37% HCl was added to stop the alkaline treatment by adjusting the pH value to 7.0. DNA and RNA were cleaved by DNase I and RNase A at 37 °C for 1 h, following digestion with proteinase K overnight. Finally, PAR was precipitated by phenol-chloroform-isoamyl alcohol. For dot blotting assay, the PAR was spotted onto the nitrocellulose membrane and subjected to western blot using anti-PAR antibody. For the total level of PAR detection, the PAR sample was examined at 259 nm using UV-spectrophotometer. For PAR chain components examination, the product was subjected into LC–MS/MS.

**LC–MS/MS**. Due to the chain length and complexity of PAR is variable, the PAR was digested to single nucleosides by pyrophosphatase (PPase) and alkaline phosphatase (AP) prior to LC–MS/MS detection[29]. Three compositions of the digestion products are adenosine (Ado), ribosyladenosine (R-Ado) and diribosyladenosine (R$_2$-Ado), which represent the terminal, linear, or branched PAR, respectively. Liquid chromatography (HPLC) for all LC–MS/MS runs were performed on Agilent 6520B QTOF mass spectrometer equipment with analytic column (Phenomenex, size 150 × 2 × 3 micro) at a maximum pressure of 300 bars. Solution A for HPLC was 0.1% formic acid in water and Solution B for HPLC was acetonitrile. The most dominant daughter ion of R-Ado (m/z of 400) after collision was adenine (m/z of 136). Therefore the transitions of m/z 400 → 136 and m/z 532 → 136 were monitored for the quantification of R-Ado and R$_2$-Ado, respectively. The ratio of Ado, R-Ado and R$_2$-Ado were quantified by their respective peak areas. The branch frequency was calculated by R$_2$-Ado/R-Ado.

For the APLF recognized branched PAR chain detection, the PAR was incubated with the beads-coupled GST-tagged APLF PBZ protein for 2 h at 4 °C. Then the PAR-protein complex was partially digested by PARG catalytic truncated protein in PBS buffer for 10 min at room temperature. The APLF PBZ-bound residues were eluted and digested by PPase and AP prior to LC–MS/MS. Meantime, purified GST protein and GST-tagged RNF146 WWE protein were used as negative and positive controls under the same tested condition.

**Inducible CRISPR/Cas9 platform to generate solo DNA double-strand break**. To produce Cas9 lentivirus, pCW-Cas9 was co-transfected into 293T cells along with packaging (psPAX2) and envelope (pMD2.G) using Lipofectamine 2000 (Invitrogen). The lentivirus-containing media were collected 48 h post-transfection and filtered to remove cells. The lentiviral particles were used to infect the cells for 48 h. To generate the U2OS-Cas9 cell line, the cells were treated with 1.5 μg/ml puromycin to select the U2OS cells stably expression of Cas9. Considering U2OS PARP2- or APLF-null cells were puromycin-resistant, we replaced the screen maker with blasticidin by modifying the pCW-Cas9 vector. A total of 5 μg/ml blasticidin was used to select the U2OS PARP2- or APLF-null cells stably expression of Cas9. The antibiotics were removed until uninfected control cells were completely killed. The Cas9 induction efficiency was tested after treatment with 1 μg/ml doxycycline (DOX) for 24 h. gRNA targeting AAVS1 was then introduced into the iCas9 cell lines to generate solo DNA double-strand break (DSB) for subsequent histone H3 removal analysis by chromatin immunoprecipitation or DNA damage repair kinetics using q-PCR.

**Immunofluorescence analysis**. To analyze γH$_2$AX foci after induction of DSB, U2OS-iCas9 cells grown on coverslips were transfected with a mock or gRNA and cultured at 37 °C for 5 h to allow DNA damage repair. Then cells were immunostained for γH$_2$AX foci, images were recorded microscopically.

To examine the PARylation on the DNA damage sites, U2OS cells were subjected to laser micro-irradiation, following 20 s, the cells was immunostained with anti-PAR antibody. The fluorescence images were captured using microscope.

To monitor the kinetic of endogenous PARP2 recruitment following DNA damage, laser micro-irradiation was performed to generate local DNA damage in U2OS cells. And the cells were immunostained with anti-PARP2 antibody. Image acquisition and analysis was carried out at different time points.

**Chromatin immunoprecipitation (ChIP)**. Cells (70–80% confluence) from two 10-cm plates were washed with PBS then cross-linked with 1% formaldehyde for 10 min at room temperature. Fixation was stopped by 200 mM glycine for 5 min. Cells were washed twice with PBS and harvested in lysis buffer (20 mM Tris-HCl, pH 8.0, 150 mM NaCl, 10 mM EDTA, 1% SDS) containing proteinase inhibitors, 1 mM DTT and 1 mM PMSF. Lysates were sonicated on ice to yield 300–1000 bp genomic DNA fragments and then centrifuged at maximal speed, 4 °C for 10 min. And the samples were diluted tenfold in dilution buffer and immunoprecipitated by anti-histone H3 antibody (2 μg) with 30 μl protein G beads over night at 4 °C. IgG as negative control was also used. The beads were collected and washed extensively. The immuno-complex was eluted with freshly prepared elution buffer (1% SDS, 100 mM NaCHO$_3$) for 30 min at room temperature. Crosslinks were reversed with an incubation of the samples with 300 mM NaCl and proteinase K at 65 °C for 4 h. Immunoprecipitated DNA was purified using the QIAquick PCR Purification Kit. Two microliters of the DNA sample was subjected into the q-PCR reactions. The

corresponding primers are listed in Supplementary Table 3. The results were normalized for the signal of the input and were expressed as a percentage of the signal with the antibody.

**Live-cell imaging by laser micro-irradiation.** GFP-tagged constructs were transfected into the indicated cells which were plated on 35-mm glass bottom dishes. Cellular DNA damages were generated in the nuclei of cultured cells by micro-irradiation with a pulsed nitrogen laser (Spectra-Physics; 365 nm, 10 Hz pulse). The laser system was directly coupled to the epifluorescence path of the microscope for time-lapse imaging and focused through a Plan-Apochromat ×63/NA 1.40 oil immersion objective. The output of the laser power was set at 50–70% of the maximum as indicated. The green fluorescence strips were recorded at indicated time points and then analyzed with Image J software. All results represent images of 20 cells from three independent experiments.

**HR and NHEJ assays.** To measure the repair frequency, DR-GFP U2OS or EJ5-GFP U2OS cells were plated into six-well plates and transfected the next day with the indicated siRNAs and pCBASceI vector using Lipofectamine 2000 (Invitrogen)[60]. Forty-eight hours later, cells were harvested and washed with PBS, and the GFP positive cells were recorded by flow cytometry on a FACSCalibur.

**Neutral comet assay.** For evaluating DNA double-strand breaks, the neutral version of comet assay was performed. Briefly, cells were exposed to 5 Gy and recovered at different time points. Then the cells were collected and mixed with 0.8% low melting agarose and layered onto agarose-coated slides. Slides were then submerged into cold lysis buffer for 3 h. After lysis, slides were incubated for 1 h in electrophoresis buffer. After electrophoresis, slides were neutralized, placed into 100% ethanol and then air-dried. Slides were subsequently stained with 5 μg/ml propidium iodide and images were taken using a fluorescent microscope (Olympus). Average Olive Tail Moment (OTM) was analyzed (50 cells/slide) by using Comet Assay Software Project Casp-1.2.2 (University of Wroclaw, Poland). All experiments were repeated three times.

**Statistical analysis.** Data are represented as mean ± s.d. as indicated from three independent experiments. Significance of differences was evaluated by Student's $t$-test. NS: non significant; *statistically significant ($p < 0.05$). **statistically significant ($p < 0.01$). ***statistically significant ($p < 0.001$).

**Data availability.** All uncropped blots and gels are shown in Supplementary Figure 11 and all data are available from the corresponding author upon reasonable request.

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

## Acknowledgements

The mass spectrometry was performed with equipment at the City of Hope Mass Spectrometry and Proteomics Core Facility. We are grateful to Roger Moore for providing technical support on the mass spectrometry analysis. We also thank Dr. John Pascal for His-PARP1 construct. We appreciate Dr. Jeremy Stark generous sharing of DR-GFP U2OS and EJ5-GFP U2OS cell lines. This work was supported in part by grants from National Institutes of Health (CA132755, CA130899 and CA187209 to X.Y.) and the Department of Defense (BA160420 to X.Y.). X.Y. is a recipient of Research Scholar Award from Leukemia and Lymphoma Society.

## Author contributions

X.Y. designed the whole project. Q.C. performed the experiments and created the figures. M.A.K. and F.D. participated in the project with technical supports. Q.C. and X.Y. analyzed the data. Q.C., M.A.K., and X.Y. wrote the manuscript. All authors reviewed the results and approved the final version of the manuscript.

## Additional information

**Competing interests:** The authors declare no competing interests.

