## [Peer Review File · Nature Communications]

Reviewers' Comments:

Reviewer #1:

Remarks to the Author:

In this manuscript, Chen and colleagues investigated the regulation and function of PARP2. Using a series of assays they established, the authors discovered that PARP2 has a significant role in the formation for branched PAR chains. Mechanistically, the authors showed that PARP1-mediated PARylation is recognized by the N-terminus of PARP2, which is responsible for PARP2 recruitment and branched PAR formation. Moreover, the authors demonstrated that the branched PAR in turn is recognized by APLF and contributes to histone removal following DNA damage.

This is a well-executed study with two interesting discoveries. The first is that PARP2 is mainly involved in the formation of branched PAR chains. The second is that branched PAR chains recruit APLF to promote histone removal during DNA repair. Both of these are novel findings that warrant publication.

The authors showed nicely that PARP2 is activated not only by DNA but also by PAR. It will be interesting to determine whether or not PARP1 also behaves similarly. In addition, the authors should also examine DSB repair in the absence of PARP2 or APLF to reveal whether or not these two proteins indeed act in the same pathway.

Reviewer #2:

Remarks to the Author:

In this manuscript, Chen and colleagues address the effects of PARP2 to discover that PARP2 produces branching in PAR chains when activated by PAR. Furthermore they show that wild-type APLF preferentially binds branched PAR and that the lack of PARP2 or APLF compromises DNA damage repair and histone mobilization in a cellular Cas9-based DNA damage assay. The physiological relevance of PAR branching has been a long-standing question in the field. The manuscript is easy to read and the figures are clear. The presented findings are very intriguing. Most puzzling is the essentially identical phenotype of APLF and PARP2 knockouts. One might expect that the DNA repair phenotype of PARP2 KO (Fig 6) cells would be milder than that of the APLF KO cells if it goes through APLF recruitment to the DNA damage through binding branched PAR. APLF recruitment is still observed in PARP2 KO cells (Fig 5D). Is DNA repair so sensitive to the reduction of APLF levels or function? If PARP2 ADP-ribosylated a specific substrate (in theory it could be even APLF) crucial for APLF activation the phenotype could be the same irrespective of PAR branching, couldn't it be?

Could the percentage of branching in a typical PAR chain +/- PARP2 be estimated?

Based on the relative levels shown in Fig 1, the lack of PARP2 only reduces branching 50%. What is the other source? PARP1? Could PAR branching be quantified for in vitro PARP1 produced PAR?

In Fig 4, could PARP2 and NTR recruitment be repeated in the presence of PARP inhibitor? This would further strengthen that PARP2 is recruited to the PARP1 produced PAR.

Authors show changes in relative H3 upon DNA damage in Fig 6. What are the absolute H3 levels at these loci in the WT and KO cells? Some of the differences could stem from already depleted H3/ fewer nucleosomes in the KO cells. How long is the targeting sgRNA present in the cells? How efficiently is the site re-cut by Cas9? If APLF facilitates NHEJ, APLF KO cells might repair the cut with HR more often therefore facilitating re-cutting (NHEJ resulting in more mutated therefore less efficiently cut DNA), which could also explain Fig 6F. Could authors address if the repair defects is through defective NHEJ, HR or both? Can the repair defect observed in the KO cells be rescued by reintroducing WT PARP2 or APLF (similar to Suppl Fig S7)? Were the used cells synchronized? Is the cell-cycle distribution of the used cell lines essentially the same?

Reviewer #3:

Remarks to the Author:

In this manuscript by Chen et al., the authors have examined the activation of PARP2 and its role in the formation branched poly(ADP-ribose) chains in response to DNA damage. Using PAPR2 deficient MEFs and U2OS cells and a series of elegant experiments, the authors show that loss of PARP2 does not have a significant effect on overall parylation, instead it resulted in significant reduction of branched PAR chains. In contrast, PARP1 loss reduces overall PAR levels. The authors also show activation of PARP2 by PAR chains. Using an in vitro parylation assay and mass spectrometry, branched parylation was shown to be markedly increased in the presence of PAR chains and PAPR2 but not PAPR1, suggesting that PARP2 and not PAPR1 is capable of generating branched PAR chains. Furthermore, the N-terminal region of PAPR2 was shown to be critical for parylation by PAPR2. The authors also examined the kinetics of PARP1 and PAPR2 recruitment to the site of DNA damage and found PARP1 to be recruited before PAPR2 and the recruitment of the latter to be dependent on catalytic activity of PARP1. The functional significance of branched parylation was revealed by its role in recruitment of APLF, which facilitates chromatin remodeling by removal of histone H3 at the site of DNA damage.

Overall, the findings are very interesting and important and show a clear functional difference between PARP1 and PARP2 in terms of the PAR chains they generate and the kinetics of their recruitment. Experimental approaches used in the study are appropriate and the results are very convincing. In a study by Sukhanova et al. (2015 NAR 44(6):e60), using atomic force microscope imaging, it was shown that both PARP1 and PAPR2 are capable of branched parylation. To explore this possibility in vivo, have the authors examined the effect of PARP1 overexpression in PAPR2^{-/-} cells. It will be important to find out if in the absence of PARP2, can higher PAPR1 levels synthesize branched PAR chains?

Some minor comments:

1. In several initial experiments, PARP1^{-/-} control was not included, e.g. Fig 1B, 1D, 1E.
2. Data shown in Suppl. Figure S1 should be included in Fig 1C.
3. Similarly, data shown in Suppl. Figure S4 should be included in Figure 2.
4. Figure 4E, authors should show that PARP1 E988A recruitment to the site of DNA damage is not affected.
5. Include p value in Figure 1C
6. References 38 and 42 are the same.

Reviewer #4:

Remarks to the Author:

PARylation plays a critical role in regulating many aspects of cell stress responses. Even though the existence of branched PARylation has been demonstrated more than 35 years ago, the regulation and function of this intriguing topological feature within PAR polymers remain elusive. In this very exciting paper, Chen et al., developed a clever method to identify and quantify branched PAR structures, and showed that PARP2 is the "writer" for branched PAR polymers. Specifically, they demonstrated that PARP2 is activated by binding to preformed linear PAR chains, thereby catalyzing the formation of a second wave of PARylation (including the branched PAR). Furthermore, they identified tandem PBZ domains as the "reader" of branched PAR. Finally, such domains in APLF mediate the branched PAR-dependent recruitment of APLF during genotoxic stress, and subsequent histone eviction. Overall, this is a well presented manuscript that is of profound implications. The methodological aspects are novel; the experimental design and result interpretation are robust and comprehensive. I had just a few things that I would like to see amended prior to publication.

Specific comments:

(1) Page 4, "expend" should be "expand".

(2) Page 5, why there is also a decrease in the Ado levels in the PARP2 KO cells?

(3) Page 6, based on the data, can the authors estimate the percentage of branching points, in the total PAR polymers, and how this number is changed upon PARP2 KO. R2-Ado levels decrease by about 50% in PARP2 KO cells. What is contributing to formation of the remaining branched PAR?

(4) Page 8, the authors need to discuss the potential structural differences in the catalytic domain between PARP1 and PARP2 that lead to the formation of linear vs. branched PAR.

(5) Page 10, what is the difference between the two PBZ domains that leads to the recognition of the two vs. one ADP-ribose unit?

(6) Page 16, "difference" should be "differences". There are a few other typos in the manuscript.

Response to reviewers' comments:

We are very grateful to the constructive suggestions from the four reviewers. Following the reviewers' suggestions, we have performed additional experiments and modified our manuscript. As listed below, we have point-by-point addressed all the concerns raised from all four reviewers.

Reviewer #1

General comments:

“In this manuscript, Chen and colleagues investigated the regulation and function of PARP2. Using a series of assays they established, the authors discovered that PARP2 has a significant role in the formation for branched PAR chains... This is a well-executed study with two interesting discoveries. The first is that PARP2 is mainly involved in the formation of branched PAR chains. The second is that branched PAR chains recruit APLF to promote histone removal during DNA repair. Both of these are novel findings that warrant publication.”

Thank you for the support.

Minor concerns:

1. *“The authors showed nicely that PARP2 is activated not only by DNA but also by PAR. It will be interesting to determine whether or not PARP1 also behaves similarly.”*

Answer: As suggested by the reviewer, we have examined whether or not PARP1 can be activated by PAR using the same method. Our results show that PARP1 cannot be activated by PAR. The data were included in the revised Supplemental Figure S4B.

2. *“In addition, the authors should also examine DSB repair in the absence of PARP2 or APLF to reveal whether or not these two proteins indeed act in the same pathway.”*

Answer: As suggested by the reviewer, we treated PARP2 or APLF-deficient U2OS cells with IR, and measured the DSB repair using neutral comet assays. We found that lacking either PARP2 or APLF impairs DSB repair. The results have included in the revised Supplemental Figure S9A.

Reviewer #2

General comments:

“In this manuscript, Chen and colleagues address the effects of PARP2 to discover that PARP2 produces branching in PAR chains when activated by PAR... The physiological relevance of PAR branching has been a long-standing question in the field. The manuscript is easy to read and the figures are clear. The presented findings are very intriguing.”

Thank you for the positive comment.

Specific concerns:

1. *“Most puzzling is the essentially identical phenotype of APLF and PARP2 knockouts. One might expect that the DNA repair phenotype of PARP2 KO (Fig 6) cells would be milder than that of the APLF KOs if it goes through APLF recruitment to the DNA damage through binding branched PAR. APLF recruitment is still observed in PARP2 KOs (Fig 5D). Is DNA repair so sensitive to the reduction of APLF levels or function? If PARP2 ADP-ribosylated a specific substrate (in theory it could be even APLF) crucial for APLF activation the phenotype could be the same irrespective of PAR branching, couldn't it be?”*

Answer: We agree with the reviewer that similar repair defect phenotype observed in APLF and PARP2-deficient cells could be complicated. To further confirm the cellular phenomenon, we treated APLF and PARP2-deficient cells with IR and used neutral comet assays to examine the DSB repair. Again, we found that the DSB repair kinetics in APLF and PARP2-deficient cells were similar (Supplemental Figure S9A). The results are consistent with previous publications that both APLF and PARP2 are important for DSB repair¹⁻⁶. As the authors suggested, theoretically, the repair defects in PARP2-deficient cells should be milder than that in APLF-deficient cells, because lacking PARP2 only reduces more than 50 % of branched chains, and a set of APLF was still able to be recruited to the sites of DNA damage. However, it is possible that besides APLF, other repair factors may also recognize the PARP2-dependent branched chain. Lacking PARP2-dependent branched chain may also impair the recruitments of other branched chain readers. In other words, APLF and PARP2 have overlapping but also independent functions in DNA damage repair.

It is also possible that PARP2 ADP-ribosylates one substrate for the recruitment of APLF. But our results suggest that the PBZ motif of APLF specifically recognizes the branched chain formation (Figure 5). Moreover, loss of the PBZ motif of APLF abolishes the function of APLF in DNA damage repair. Thus, it is likely that the activation of APLF relies on the interaction between the PBZ and the branched PAR chain. However, to be cautious on data interpretation, we discussed the possibilities that the reviewer has mentioned (Page 19 Line 9-22). Further analysis on APLF-dependent pathway and identification of the PARP2 substrates will elucidate the detailed molecular mechanism.

2. *“Could the percentage of branching in a typical PAR chain +/- PARP2 be estimated?”*

Answer: Following the suggestion, we quantitatively measured the branched chain (R_2 -Ado) in both wild type and PARP2 null cells with LC-MS/MS. Based on a previously demonstrated approach⁷, we compared the ratio between R_2 -Ado and R-Ado. We understand that this method may only allow us to estimate the percentage of branching site, and it may not be very accurate because we do not have radio-isotope labeled R_2 -Ado as an internal reference. However, it is the only available approach for us to have an estimated branching site. Here, we found that the branched unit is ~2 % of the linear chain unit in the wild type cells. In PARP2 null cells, the branched site is less than 1 % of the linear chain unit. The results were included in Supplemental Figure S3.

3. *“Based on the relative levels shown in Fig 1, the lack of PARP2 only reduces branching 50%.”*

What is the other source? PARP1? Could PAR branching be quantified for in vitro PARP1 produced PAR?”

Answer: This is a very good question. It is likely that other PARPs mediate the remaining branched chain. As suggested, we measured the ratio between R₂-Ado and R-Ado the *in-vitro* PARylation assays. With only PARP1, we found that the branched site is ~ 1 % of the linear unit when only PARP1 was used to catalyze the PARylation (Supplemental Figure S5A). When we added PARP2, we found that branched site increased ~ 2-fold (Figure 2B). The results suggest that PARP1 is also able to catalyze branched chain, but at lower rate. Moreover, other PARPs, such as PARP3 and PARP10, also participate in DNA damage repair⁸⁻¹⁰. It is possible that other PARPs catalyze single or oligo branch units on top of the existed PAR chains. Based on these data, we also included a short discussion in the revised text (Page 8 Line 16-18).

4. “In Fig 4, could PARP2 and NTR recruitment be repeated in the presence of PARP inhibitor? This would further strengthen that PARP2 is recruited to the PARP1 produced PAR.”

Answer: Following the suggestion, we expressed the PARP2 and NTR in U2OS cells. The cells were pre-treated with olaparib and followed by laser micro-irradiation. We found that olaparib treatment suppressed the recruitment of PARP2 and NTR (Supplemental Figure S6A). Notably, current PARP inhibitors suppressed the enzymatic activities of both PARP1 and PARP2.

5. “Authors show changes in relative H3 upon DNA damage in Fig 6. What are the absolute H3 levels at these loci in the WT and KO cells? Some of the differences could stem from already depleted H3/fewer nucleosomes in the KO cells. Before cutting, the H3 level in the WT and PARP2 KO cells.”

Answer: As suggested by the reviewer, we examined the histone H3 levels at those loci before Cas9-induced cutting. We performed ChIP assays and q-PCR. The Cq values from q-PCR were included in Supplemental Figure S8A to reflect the absolute levels of H3. The results show that the levels of H3 were not changed in the WT and KO cells.

6. “How long is the targeting sgRNA present in the cells? How efficiently is the site re-cut by Cas9?”

Answer: We agree with the review that Cas9 may recut the targeting site. However, with the current system, we could not estimate the recutting efficiency due to heterogeneous cutting time. Due to efficacy of CRISPR/Cas9, we only detected that the DSB is generated in ~ 60 % of the cells. Moreover, the lesions have been largely repaired in four hours. Thus, we estimate that the recutting events could be low. Previous studies have been that half-life of sgRNA could be as short as 15 minutes¹¹. Thus, the recutting-induced by Cas9 could be negligible in the current analyses.

7. *“If APLF facilitates NHEJ, APLF KO cells might repair the cut with HR more often therefore facilitating re-cutting (NHEJ resulting in more mutated therefore less efficiently cut DNA), which could also explain Fig 6F. Could authors address if the repair defects is through defective NHEJ, HR or both? Can the repair defect observed in the KO cells be rescued by reintroducing WT PARP2 or APLF (similar to Suppl Fig S7)? ”*

Answer: Thanks for the constructive suggestions. To examine if APLF is involved in HR and NHEJ, we depleted PARP2 or APLF in DR-GFP U2OS and EJ5-GFP U2OS cells respectively. Based on these GFP reporter assay, we found that NHEJ was clearly impaired when cells lost PARP2 or APLF. However, HR was also mildly suppressed when cells were lacking PARP2 or APLF (Supplemental Figure S9B). The results are in agreement with previous studies on PARP2 and APLF^{1,5,12,13}. Thus, the detailed repair mechanism could be complicated. Here, we only examined the histone removal as nucleosomal histone is a barrier for any type of DSB repair. As suggested by the reviewer, we also reintroduced full length of PARP2 and APLF to rescue the repair defects in the KO cells. The results exclude the off-target effect, and have been included Supplemental Figure S8B.

8. *“Were the used cells synchronized? Is the cell-cycle distribution of the used cell lines essentially the same?”*

Answer: We used asynchronous cells in the repair assays. As APLF is able to recognize the branching site, the purpose for these assays is to demonstrate the overlapping function of PARP2 and APLF in DNA damage repair. Moreover, we have shown that the branching binding motif of APLF plays an important role in the repair as well (Supplemental Figure S8C). We agree with the reviewer that cell cycle analysis may flush out novel molecular mechanism on PARP2 or APLF-dependent repair, which is not the major focus of this study. Moreover, in the current system, we have to deliver sgRNA and induce the expression of Cas9, and then observe the repair kinetics and histone removal. With another layer of synchronizing cells such as double thymidine block and release, it may become much more complicated, and many other assays should be performed to elucidate the underlying mechanism if cell cycle regulates the PARP2 or APLF-dependent repair. We wish to develop a much simplified assay system to examine such molecular mechanism in future.

Reviewer #3

General comments:

“In this manuscript by Chen et al., the authors have examined the activation of PARP2 and its role in the formation branched poly(ADP-ribose) chains in response to DNA damage ... Overall, the findings are very interesting and important and show a clear functional difference between PARP1 and PARP2 in terms of the PAR chains they generate and the kinetics of their recruitment. Experimental approaches used in the study are appropriate and the results are very convincing.”

Thank you for the generous comments and support!

1. *“In a study by Sukhanova et al. (2015 NAR 44(6):e60), using atomic force microscope imaging, it was shown that both PARP1 and PARP2 are capable of branched parylation. To explore this possibility in vivo, have the authors examined the effect of PARP1 overexpression in PARP2^{-/-} cells. It will be important to find out if in the absence of PARP2, can higher PARP1 levels synthesise branched PAR chains?”*

Answer: Yes, we agree with reviewer and the previous study that both PARP1 and PARP2 are able to synthesize branched PAR chain. Following the suggestion, we over-expressed PARP1 in the PARP2 KO cells. However, we found that both R-Ado and R₂-Ado were increased (Supplemental Figure S5B), suggesting that overexpression of PARP1 induced additional PARylation. We also examined PARP1-dependent PARylation in vitro. At least in vitro, PAR is not able to activate PARP1 (Supplemental Figure S4B). Moreover, we are able to detect the branched site in the PARP1-dependent PARylation in vitro, suggest that PARP1 is able to synthesize branched PAR chain (Supplemental Figure S5A). However, when we added additional PARP2, more branched units were detected, suggesting that PARP2 can be activated by PARP1-induced PAR chain (Figure 2B).

Some minor comments:

1. *“In several initial experiments, PARP1^{-/-} control was not included, e.g. Fig 1B, 1D, 1E.”*

Answer: Following the suggestion, the PARP1-deficient cell control was included in Supplemental Figure S1 and Fig 1C. However, loss of PARP1 abolishes ~ 90 % PAR synthesis. Thus, the endogenous PAR level, especially R₂-Ado, is too low to be detected in the PARP1^{-/-} cells.

2. *“Data shown in Suppl. Figure S1 should be included in Fig 1C.”*

Answer: We have moved Suppl. Figure S1 into revised Fig 1C.

3. *“Similarly, data shown in Suppl. Figure S4 should be included in Figure 2.”*

Answer: We have moved Suppl. Figure S4 into revised Fig 2.

4. *“Figure 4E, authors should show that PARP1 E988A recruitment to the site of DNA damage is not affected.”*

Answer: Following the suggestion, we examined the recruitment of the E988A mutant. The recruitment kinetics was included in Supplemental Figure S6C.

5. *“Include p value in Figure 1C.”*

Answer: Thank you for the reminder. We have added the statistical analyses and p value in the revised figures and figure legends.

6. *“References 38 and 42 are the same.”*

Answer: This error has been corrected in the revised manuscript.

Reviewer #4

General comments:

“PARylation plays a critical role in regulating many aspects of cell stress responses. Even though the existence of branched PARylation has been demonstrated more than 35 years ago, the regulation and function of this intriguing topological feature within PAR polymers remain elusive. In this very exciting paper, Chen et al., developed a clever method to identify and quantify branched PAR structures, and showed that PARP2 is the “writer” for branched PAR polymers ... this is a well presented manuscript that is of profound implications. The methodological aspects are novel; the experimental design and result interpretation are robust and comprehensive. I had just a few things that I would like to see amended prior to publication.”

Thank you for the generous comments and support!

Specific comments:

(1) Page 4, “expend” should be “expand”.

Answer: The typo has been corrected.

(2) Page 5, why there is also a decrease in the Ado levels in the PARP2 KO cells?

Answer: This is a very good question. It is possible that PARP2-dependent PAR chain is relatively short. Loss of PARP2 will also reduce a significant amount of terminal units at the branched chains. In other words, loss of branched chains will also lose the terminal units.

(3) Page 6, based on the data, can the authors estimate the percentage of branching points, in the total PAR polymers, and how this number is changed upon PARP2 KO. R₂-Ado levels decrease by about 50% in PARP2 KO cells. What is contributing to formation of the remaining branched PAR?

Answer: This is another very good question, which is also asked by Reviewer #2 (Question #3). Based on a previously demonstrated approach⁷, we compared the ratio between R₂-Ado and R-Ado. We understand that this method may only allow us to estimate the percentage of branching site, and it may not be very accurate because we do not have radio-isotope labeled R₂-Ado as an

internal reference. However, it is the only available approach for us to have an estimated branching site. Here, we found that the branched unit is ~ 2 % of the linear chain unit in the wild type cells. In PARP2 null cells, the branched site is less than 1 % of the linear chain unit. The results were included in Supplemental Figure S3.

Moreover, other PARPs mediate the remaining branched chain. As suggested, we measured the ratio between R₂-Ado and R-Ado the *in-vitro* PARylation assays. With only PARP1, we found that the branched site is ~ 1 % of the linear unit when only PARP1 was used to catalyze the PARylation (Supplemental Figure S5A). When we added PARP2, we found that branched site increased ~ 2-fold (Figure 2B). The results suggest that PARP1 is also able to catalyze branched chain, but at lower rate. Moreover, other PARPs, such as PARP3 and PARP10⁸⁻¹⁰, also participate in DNA damage repair. It is possible that other PARPs catalyze single or oligo branch units on top of the existed PAR chains. Based on these data, we also included a short discussion in the revised text (Page 8 Line 16-18).

(4) Page 8, the authors need to discuss the potential structural differences in the catalytic domain between PARP1 and PARP2 that lead to the formation of linear vs. branched PAR.

Answer: The structures of the catalytic pockets of PARP1 and PARP2 have been examined¹⁴. In fact, the detailed structures are quite different in PARP1 and PARP2. The catalytic pocket can be divided into acceptor site and donor site. Although the binding sites of NAD⁺ donor are quite similar, the substrate acceptor sites are quite different. Compared to that in PARP1, PARP2 has a unique extended loop with six additional residues (Leu523-Thr529) in the acceptor pocket that changes the tertiary structure. It is very likely that the different orientation of substrate acceptor pocket determine the linear and branched PAR chain formation. We have included this part of discuss in Page 17 line 10-19.

(5) Page 10, what is the difference between the two PBZ domains that leads to the recognition of the two vs. one ADP-ribose unit?

Answer: The structure of PBZ motif has been characterized¹⁵. The tertiary structure of the PBZ1 of APLF is very similar to that of CHFR. The structural analysis shows that the PBZ of CHFR contains two ADPr-binding sites. The key residues are also conserved in the PBZ1 of APLF. However, in the PBZ2 of APLF, a proline residue replaces the methionine residue in PBZ1, and blocks the first ADPR-binding site. Consistently, both CHFR PBZ and APLF PBZ1, but not APLF PBZ2, have higher affinity with PAR. We comment this difference in the revised Page 18 line 1-3.

(6) Page 16, “difference” should be “differences”. There are a few other typos in the manuscript.

Answer: Typos have been corrected by additional proofreading.

References:

- 1 Beck, C., Robert, I., Reina-San-Martin, B., Schreiber, V. & Dantzer, F. Poly(ADP-ribose) polymerases in double-strand break repair: focus on PARP1, PARP2 and PARP3. *Exp Cell Res* **329**, 18-25, doi:10.1016/j.yexcr.2014.07.003 (2014).
- 2 Bekker-Jensen, S. *et al.* Human Xip1 (C2orf13) is a novel regulator of cellular responses to DNA strand breaks. *J Biol Chem* **282**, 19638-19643, doi:10.1074/jbc.C700060200 (2007).
- 3 Chalmers, A., Johnston, P., Woodcock, M., Joiner, M. & Marples, B. PARP-1, PARP-2, and the cellular response to low doses of ionizing radiation. *Int J Radiat Oncol Biol Phys* **58**, 410-419 (2004).
- 4 Liu, C., Vyas, A., Kassab, M. A., Singh, A. K. & Yu, X. The role of poly ADP-ribosylation in the first wave of DNA damage response. *Nucleic Acids Res* **45**, 8129-8141, doi:10.1093/nar/gkx565 (2017).
- 5 Yelamos, J., Schreiber, V. & Dantzer, F. Toward specific functions of poly(ADP-ribose) polymerase-2. *Trends Mol Med* **14**, 169-178, doi:10.1016/j.molmed.2008.02.003 (2008).
- 6 Fouquin, A. *et al.* PARP2 controls double-strand break repair pathway choice by limiting 53BP1 accumulation at DNA damage sites and promoting end-resection. *Nucleic acids research* **45**, 12325-12339, doi:10.1093/nar/gkx881 (2017).
- 7 Martello, R., Mangerich, A., Sass, S., Dedon, P. C. & Burkle, A. Quantification of cellular poly(ADP-ribosyl)ation by stable isotope dilution mass spectrometry reveals tissue- and drug-dependent stress response dynamics. *ACS Chem Biol* **8**, 1567-1575, doi:10.1021/cb400170b (2013).
- 8 Boehler, C. *et al.* Poly(ADP-ribose) polymerase 3 (PARP3), a newcomer in cellular response to DNA damage and mitotic progression. *Proc Natl Acad Sci U S A* **108**, 2783-2788, doi:10.1073/pnas.1016574108 (2011).
- 9 Shahrour, M. A. *et al.* PARP10 deficiency manifests by severe developmental delay and DNA repair defect. *Neurogenetics* **17**, 227-232, doi:10.1007/s10048-016-0493-1 (2016).
- 10 Nicolae, C. M. *et al.* The ADP-ribosyltransferase PARP10/ARTD10 interacts with proliferating cell nuclear antigen (PCNA) and is required for DNA damage tolerance. *J Biol Chem* **289**, 13627-13637, doi:10.1074/jbc.M114.556340 (2014).
- 11 Ma, H. *et al.* CRISPR-Cas9 nuclear dynamics and target recognition in living cells. *J Cell Biol* **214**, 529-537, doi:10.1083/jcb.201604115 (2016).
- 12 Grundy, G. J. *et al.* APLF promotes the assembly and activity of non-homologous end joining protein complexes. *EMBOJ* **32**, 112-125, doi:10.1038/emboj.2012.304 (2013).
- 13 Rulten, S. L. *et al.* PARP-3 and APLF function together to accelerate nonhomologous end-joining. *Mol Cell* **41**, 33-45, doi:10.1016/j.molcel.2010.12.006 (2011).
- 14 Oliver, A. W. *et al.* Crystal structure of the catalytic fragment of murine poly(ADP-ribose) polymerase-2. *Nucleic Acids Res* **32**, 456-464, doi:10.1093/nar/gkh215 (2004).
- 15 Oberoi, J. *et al.* Structural basis of poly(ADP-ribose) recognition by the multizinc binding domain of checkpoint with forkhead-associated and RING Domains (CHFR). *J Biol Chem* **285**, 39348-39358, doi:10.1074/jbc.M110.159855 (2010).

Reviewers' Comments:

Reviewer #1:

Remarks to the Author:

The revised manuscript addressed my previous concerns.

Reviewer #2:

Remarks to the Author:

Authors have addressed my questions and the manuscript is much improved. I have no further comments.

Reviewer #3:

Remarks to the Author:

The authors have satisfactorily addressed my concerns.